# Distinct early-life mechanisms of quantity discrimination in domestic chicks

**Maria Loconsole** [iD][1]*, **Elisa Tedaldi**[1,2], **Lucia Regolin**[1]

**1** Department of General Psychology, University of Padua, Padua, Italy, **2** Department of Communication and Media Studies, University of Zurich, Zurich, Switzerland

* maria.loconsole@unipd.it

## Abstract

In research on animal numerical cognition, newly hatched domestic chicks have been shown to rely on distinct strategies when confronted with quantitative choices. In some conditions, such as after imprinting on a specific set of objects, chicks preferentially approach the larger set of familiar items, indicating sensitivity to magnitude. In other conditions, however, their responses are governed not by magnitude per se, but by the degree of similarity between the test objects and a previously experienced set, where similarity is defined in terms of conformity to specific perceptual constraints, such as the possibility of a symmetrical division into identical subsets (as in composite versus prime sets of items). In the present study, we sought to replicate both phenomena while aligning key methodological features, including the test arena, the comparison (5 vs. 9), and the age at testing. One group of chicks was imprinted on a set of identical objects to test preference for larger familiar set; another group was habituated to even-numbered sets to assess sensitivity to perceptual asymmetry in prime-numbered ones. We successfully replicated both effects: chicks preferred the larger set after imprinting and showed longer inspection of the prime-numbered set after habituation, despite its smaller magnitude. Our results show that different mechanisms supporting quantity discrimination are available from the earliest stages of life and can be triggered by task- or environment-specific factors.

## 1. Introduction

Possessing the fundamentals of numerical cognition provides strong ecological advantage [1–3] to a wide range of animal species. Even basic numerical abilities can support behaviours such as competing for territory [4], managing food resources [5], and avoiding predators [6]. An evolutionarily ancient and phylogenetically widespread system, often referred to as the "number sense", is thought to support the approximate representation of numerical quantity from early in development in humans and other species [1,7–9]. According to this view, visual numerosity is extracted rapidly and automatically, but may be influenced by continuous non-numerical cues that

**Data availability statement:** The data generated during the study are available as supplementary material S1 Dataset.

**Funding:** This research was supported by a PRIN 2017 ERC-SH4–A Grant (2017PSRHPZ) to L.R.

**Competing interests:** The authors have declared that no competing interests exist.

co-vary with number, such as cumulative surface area, total contour length, and convex hull. Formal numerical competence builds upon this system through maturation and experience, enabling more precise representations even when continuous cues are controlled [1,10–12].

Birds, in particular, have demonstrated striking competences in quantitative and numerical discrimination tasks in both laboratory and naturalistic settings [13]. Among these studies, research on domestic chicks allows us to investigate the ontogeny and early emergence of such abilities. These studies have shown that chicks can discriminate between different numerosities [14], understand ordinal relationships [15], and even solve simple arithmetic operations [16]. A well-established finding in the literature is that chicks can discriminate between groups of identical objects with which they were reared for several hours or days shortly after hatching. At test, chicks do not prefer the familiar numerosity, but instead tend to approach the larger set of familiar objects [14,17,18]. In such cases, quantitative information carries clear ecological significance, as artificially imprinted objects are perceived by the chicks as social companions. Thus, if the group is divided in two (or more) subgroups, it is advantageous for the chick to locate and re-join the larger one. For example, chicks reared with identical objects, either 3D balls [14] or 2D squares [19,20], spontaneously approached the panel occluding the larger set when presented with sets of 2vs3, 1vs4, 1vs5, 2vs4 identical objects disappearing behind separate panels. When imprinting stimuli possess stable features that animals can reliably detect, these features are integrated into a single mental representation, and the object as a whole is learned as an individual. As a consequence, when given a choice, chicks tend to orient towards the larger group of such familiar individuals.

More recent findings, however, indicate that chicks can also solve quantity discriminations using a non-mathematical strategy, and in particular they could respond to a perceptual property that is characteristic of prime-numbered sets of elements. Evidence from biology, and clinical and developmental psychology suggests that certain structural properties of sets that correspond to prime numbers may be spontaneously perceived by non-human animals [21,22] and preverbal infants [23], and in the absence of a formal numerical ability [24–26]. Specifically, sets containing a prime number of elements cannot be partitioned into smaller equal-sized subsets. Any attempt to group such sets into equal parts necessarily results in at least one subset differing in numerosity, producing a perceptually imbalanced configuration. In contrast, non-prime numerosities can be decomposed into equal groups, generating more symmetrical and perceptually balanced arrangements. While this property has been formalised in mathematical terms as non-factorisability, it originates from a structural feature of discrete sets that does not require any explicit understanding of mathematical principles, and can possibly be exploited by animals and infants without access to formal numerical concepts. In a first study, newly hatched chicks were habituated for one hour to 360 sets of bidimensional objects varying in shape, colour, and numerosity [22]. They were then tested for their preference between two novel sets: one of a novel prime numerosity and one of a novel non-prime odd numerosity (7vs.9, 9vs.11, or 13vs.15). Results showed that chicks spent more time inspecting

the prime-numbered set, regardless of whether it was the smaller or larger of the pair. Under the proposed framework, chicks were able to discriminate prime and non-prime numerosities because the first resist symmetrical grouping, leading to perceptually irregular patterns. The preference observed in these paradigms is therefore interpreted as a response to the asymmetry or novelty generated by non-factorisable groupings. Exposure to continuously varying stimuli prevents chicks from learning the individual features of each set and instead promotes the extraction of a more general regularity shared across experiences. In this scenario, chicks are less likely to respond to the larger set, as observed with imprinted objects, and are more likely to respond to the abstract rule or pattern that characterises the experienced stimuli. Evidence for a similar mechanism has been further supported by recent data from 8-month-old infants, who displayed a comparable preference for prime sets after habituation to even numerosities [23], suggesting that a shared mechanism may underpin early quantity discrimination across species.

Notably, this study also provided the earliest behavioural evidence of quantity discrimination in this species, as the entire experimental procedure was conducted on the day of hatching. This contrasts with most chick cognition studies, including those investigating imprinting-based preferences between different quantities, which typically test animals at 3–4 days of age [14,17,18,27]. Beyond the age of testing, the two lines of research differ in several important methodological aspects. In imprinting studies, the standard measure is whether chicks choose the larger set across repeated trials, yielding a binary response that is then used to calculate a percentage of correct choices. In contrast, the perceptual symmetry study assessed the amount of time chicks spent inspecting each set, tapping into a mechanism based on sustained attention and exploratory behaviour, rather than discrete decision-making. Furthermore, imprinting studies usually involve relatively small and easily discriminable quantities (e.g., 1vs2, 2vs3), whereas the perceptual grouping study examined larger and more challenging comparisons (e.g., 7vs9, 9vs11).

In the present study, we aimed to replicate both effects using two parallel paradigms designed to minimize the methodological differences between the two research lines. Specifically, chicks were tested at the same age (24 hours after hatching), presented with the same quantitative comparison (5 vs. 9), using the same behavioural measure (time spent exploring each set) as a proxy of their preference for either set. Our results successfully replicated the original findings: following imprinting, chicks preferred the larger set, demonstrating that quantity discrimination is evident at a very early developmental stage. Following habituation, chicks spent more time inspecting the prime set, despite it being the smaller set, providing a replication of perceptual grouping effects and further supporting previous evidence from both birds and infants.

## 2. Materials and methods

### 2.1. Subjects

Fertilized eggs of domestic chicken (*Gallus gallus*) were obtained from a local hatchery (Incubatoio La Pellegrina, San Pietro in Gu, Padova, IT) and incubated at controlled temperature (37.5°C) and humidity (55–66%). Upon hatching subjects were sexed by the wing and randomly assigned to one of two possible experimental groups (for each group N = 35). For the imprinting procedure (Group 1) we selected female subjects, as they show stronger social attachment responses and are more motivated to social rewards [28,29] and in line with previous imprinting studies [14,17]. For the habituation procedure (Group 2) we used both male and female chicks as no sex-related differences were reported for this procedure [22]. Both groups of chicks were tested at the same age, i.e., about 24 hours after hatching and, at the end of the experimental procedures, animals were donated to local farmers.

The experiments complied with all applicable national and European laws concerning the use of animals in research and were approved by the Italian National Ministry of Health (N.I.H.). All procedures employed in the experiments included in this study were approved by the Italian Ministry of Health (permit number: 196/2017-PR granted on 24/02/2017).

## 2.2. Imprinting procedure

**2.2.1. Rearing conditions.** We tested a first group of 35 female one-day-old domestic chicks (all females). Upon hatching, chicks were singly placed in standard metal cages (28x32x40 cm) with 7 identical red plastic balls (4x3x3 cm) suspended at approximately 3 cm from the cage floor, via a transparent thread. Food and water were available *ad libitum* in the cage. A 36W lamp was placed 45 cm above the cage floor and provided light according to a temporized light-dark cycle (light from 7am to 7 pm, followed by dark-light alternated phases of 2/3h each).

Chicks remained in their home cages together with the artificial imprinting objects for 24h after hatching, and then they entered the test. This procedure is known to trigger a strong social attachment, for which chicks become highly motivated to re-join the larger group of these objects [14,19,27]. One hour before testing, chicks were food deprived, while water remained constantly available.

**2.2.2. Test.** At test (Fig 1), chicks were individually placed in one vertex of a triangular arena (93 cm x 62 cm x 30 cm), behind a glass partition. Opposite to the vertex were a group of 5 and a group of 9 red balls, identical to the rearing ones. Each group of objects was presented on one side (left or right), divided by a vertical partition (5x30cm). The position of the larger group was counterbalanced between subjects. After the glass partition had been removed, the chick was let free to explore the arena for 6 minutes and the amount of time spent by each group was scored.

## 2.3. Habituation procedure

**2.3.1. Habituation.** We tested a new group of 35 one-day-old chicks (26 females). Chicks were kept in the dark within the incubator to avoid any visual experience prior to habituation. About 24h post-hatching, when chicks were dry, they were individually placed in an arena identical to the one used for testing 5vs9 following imprinting, the only difference being that the side of the triangle opposite to the starting vertex consisted in a computer screen (Samsung FHD, 24", 60 Hz refresh rate). Chicks were left free to move in the arena for 1h (habituation phase, Fig 2A). During this time, sets of elements (one set at time) were presented on the monitor. Each set of elements remained visible for 10sec and was immediately replaced by the subsequent one. This way each subject saw a random combination of 360 stimuli. Each stimulus consisted in a random combination of shape (i.e., triangles, rectangles, or circles), colour (i.e., yellow, red, blue, or green) and numerosity (always even, i.e., 4, 6, 8, 10). Each element had an approximate area of 36px, whilst the spatial arrangement was random within a white area (336px) in the middle of the screen. At the end of habituation, the chick was placed singly in an empty rearing cage (same as the ones used for imprinting) for one hour before undergoing the test.

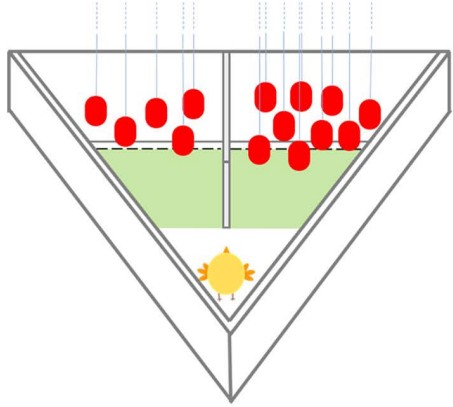 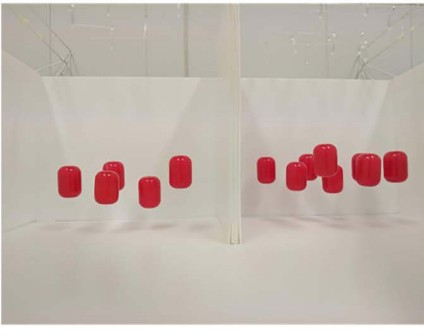

**Fig 1. Imprinting procedure.** Chicks were reared with 7 identical red balls for 24h and then tested in a 6 min free-choice task with the 5vs9 comparison. Testing stimuli were identical to the ones experienced during rearing.

**A.**

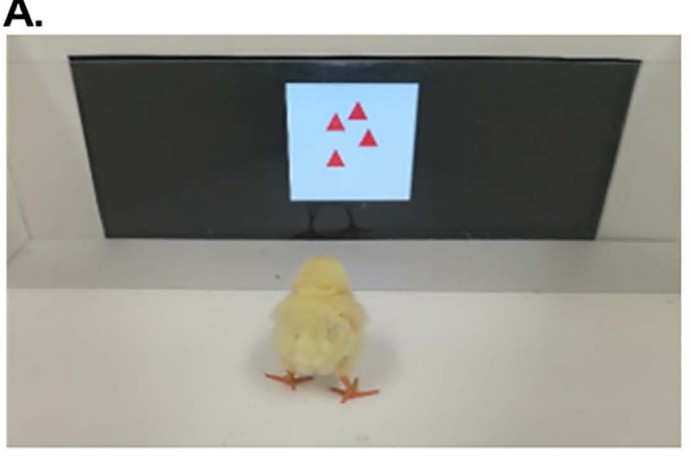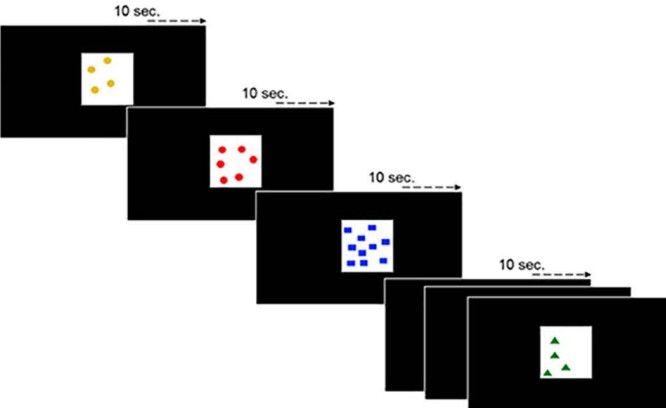

**B.**

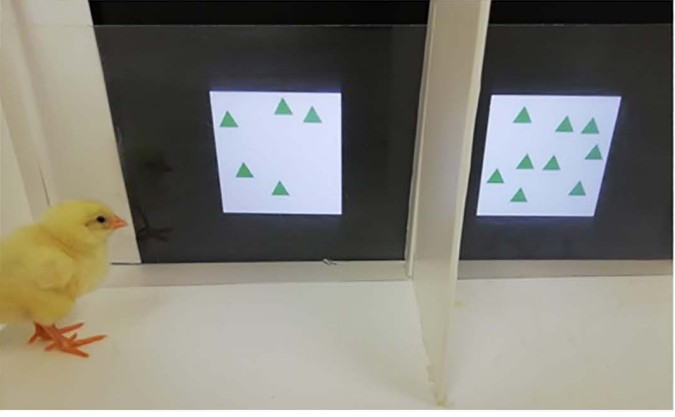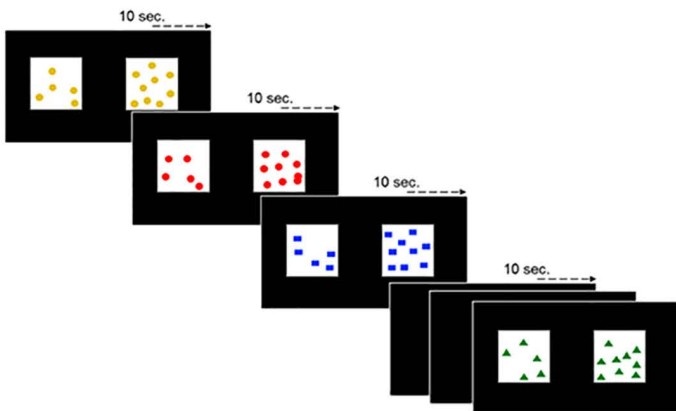

**Fig 2. Habituation procedure. A.** Chicks were exposed for 1h to the bi-dimensional (computer presented) stimuli varying in shape (circles, triangles, or squares), colour (red, yellow, blue, or green), and of even numerosity (4, 6, 10, or 12). **B.** After 1h, in which they were singly housed in a standard metal cage, chicks were tested in a 6 min free-choice task with the 5vs9 comparison.

**2.3.2. Test.** Chicks were tested in the same arena used for habituation, with the only difference that in this case two stimuli were projected at once, in the right and left halves of the screen, one of 5 elements, and one of 9 elements (Fig 2B). An opaque sagittal partition (5x30cm) divided the area in front of the screen into two choice areas. Each pair of stimuli remained visible for 10 seconds and was immediately replaced by the next one. Elements within stimuli of the same pair were of the same shape and colours. The position (left/right) of the larger group was counterbalanced between subjects. Chicks were individually placed in the vertex opposite to the monitor, behind a glass partition. After the glass partition had been removed, the chick was let free to explore the arena for 6 minutes and the amount of time spent by each stimulus was scored.

## 2.4. Data analysis

The entire testing procedures were video recorded by a camera (Canon-Legria HF-R606) placed about 30 cm above the arena. The videos were analyzed offline using BORIS [30]. In both groups, we measured the time each chick spent in proximity to each stimulus (5 and 9). Coding began when the chick crossed the vertical partition dividing the two stimulus

sets (Figs 1 and 2B) with at least one leg and the head. We interpreted this as a preference for the numerosity presented in that area, since once the chick had crossed the partition with its head, the stimulus in the opposite half was no longer visible. The videos were scored by two independent scorers, blind to the objectives of the study. Intraclass Correlation Coefficients for reliability were calculated using the R package irr [31]. We found excellent degree of reliability [32], the average ICC being 0.928 with a 95% confidence interval from 0.901 to 0.948 (F(139,140)= 26.8, p<0.0001). For data analysis we used the average of the results obtained from the two scorers.

Data were analysed using R (R 3.6.3) [33]. The dataset with raw data is available in S1 Dataset. We employed a linear mixed model with the dependent variable being the time spent in each choice area, and the independent variables being the numerosity of the comparison (i.e., 5 or 9), the pre-test procedure (i.e., imprinting or habituation), and the interaction between these two factors. Subjects were included as a random effect. Because in both procedures we recorded the same behavioural measure and employed comparable testing methods, it was possible to include the pre-test procedure as a factor in the same model, thereby allowing a direct between-group comparison of imprinting versus habituation. At the same time, the within-subject contrast between the two numerosities in each group enabled us to assess whether the expected preference effect was present, thus providing a replication of the original findings, i.e., chicks preference for the set of 9 elements (the larger) following imprinting to a set of stimuli, and the set of 5 elements (the one of a prime numerosity) following habituation to visually variable stimuli even sets. Model diagnostic were performed using the DHARMa package [34]. Simulated residuals showed no evidence of overdispersion, zero-inflation, or deviations from uniformity, suggesting that model assumptions were met. After fitting the model, we used the package emmeans [35] to obtain estimated marginal means and perform comparisons across numerosities and pre-test procedures, using Bonferroni correction. Graphs were generated using ggplot2 [36].

## 3. Results

We found no main effect of the set numerosity ($X^2$=0.021, p=0.884), indicating that chicks from the two groups did not show an overall preference for either set, nor of the pre-test procedure ($X^2$=0.048, p=0.827), indicating that overall chicks spent a similar amount of time exploring the two stimuli in the two groups. By contrast, there was a significant interaction between set numerosity and procedure ($X^2$=9.81, p=0.002, Fig 3). Chicks reared and tested with the red balls (imprinting procedure) spent longer close to the set of 9 elements (estimated mean difference (5–9) = −60.4, SE=26; t=−2.326, p=0.026), whereas chicks that underwent the habituation procedure on even numerosities (habituation procedure) spent longer close to the set of 5 elements (estimated mean difference (5–9) = 55, SE=26.2; t=2.103, p=0.043).

## 4. Discussion

In the present study we tested two separate groups of day-old chicks in the 5vs9 comparison employing two distinct experimental procedures: imprinting and habituation. These procedures were designed to elicit a preference for either the set of 9 or 5 elements, respectively, aiming at providing a direct replication of quantity discrimination effects in chicks. Importantly, the two experiments were designed to minimise procedural differences, allowing a more direct comparison. Chicks were tested at the same age, in the same arena, with the same quantity discrimination, and the same behavioural measure was recorded at test. Nevertheless, we retained a difference in the stimuli employed: Exp. 1 used 3D physical objects, whereas Exp. 2 used 2D digital projections. This choice was motivated by the need to increase objects salience and strengthen imprinting responses in Exp. 1, while allowing a rapid presentation of varying two-dimensional stimuli in Exp. 2, thereby maximising the likelihood that each procedure engaged the targeted cognitive system. Notably, the use of different stimulus formats reflects methodological considerations rather than a theoretical necessity, as both 3D and 2D stimuli have been shown to support imprinting [37,38]. Although our approach proved effective, future studies could investigate more closely matched stimulus formats to further evaluate the generalisability of these effects.

A first group of subjects underwent an imprinting procedure in which animals were exposed to a group of red balls on their first day of life. At test, chicks demonstrated a clear preference for the larger set of familiar elements in the 5vs9

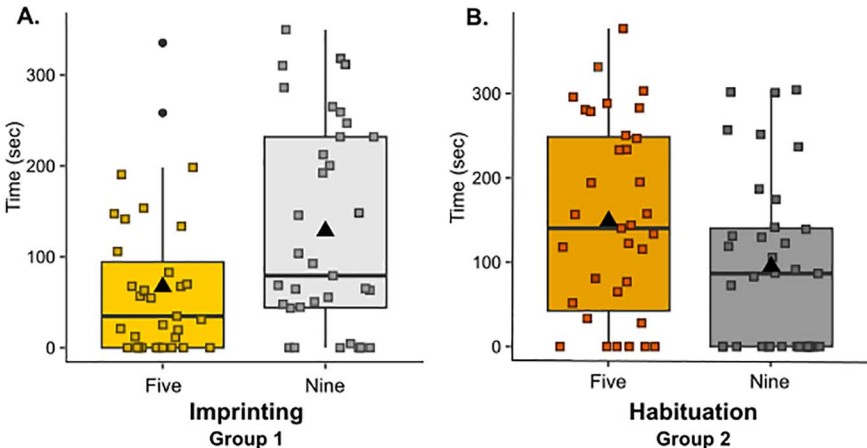

**Fig 3. Results.** On the y-axis, the time spent close to each stimulus (sec). In yellow (**A.**) and orange (**B.**) the time spent close to the numerosity 5; in light (**A.**) and dark (**B.**) grey the time spent close to the numerosity 9. Squares indicate each individual's performance; the black dots indicate outliers; the black triangle indicates the mean. **A.** Total time spent close to either set following imprinting to a set of identical objects. Chicks show a significant preference for the set of nine elements. **B.** Total time spent close to either set following habituation to visually variable sets of even numerosities. Chicks show a significant preference for the set of five elements.

comparison. This result aligns with previous studies examining quantity discrimination in 3-day and 4-day old chicks (Rugani et al. 2013, 2014, 2017). This is an important insight, as it positions quantity processing as one of the earliest cognitive abilities to manifest, potentially laying the foundation for more complex forms of cognitive processing as development progresses [7,13,16,39].

A second group of birds was habituated to sets of elements of different shapes, colours, and even numerosities. At test, subjects showed a longer inspection time for the set of five elements (the smaller set). This behaviour can be explained by a non-mathematical mechanism that enables discrimination between sets that can be grouped into symmetrical, equal-sized subgroups (i.e., sets that allow for perceptual grouping) versus sets that cannot be disassembled into such subgroups (i.e., asymmetrical sets) [22,40–42]. The successful replication of the perceptual grouping effect further strengthens the existing body of research, suggesting that this mechanism plays a pivotal role in how animals, humans included [23], may process quantitative and numerical information. By confirming these effects in newborn chicks, our study not only reinforces previous findings but also highlights the robustness and reliability of perceptual grouping as a fundamental cognitive process across species and developmental stages.

In both experiments, continuous variables covarying with numerosity were not strictly controlled, in order to more closely resemble natural conditions in which multiple magnitude-related cues remain available. The dissociation observed between experiments, with imprinting leading to a preference for the larger set and habituation eliciting a novelty-related response, suggests that distinct cognitive mechanisms were engaged by the two procedures regardless of the specific quantitative cue guiding the discrimination. It remains an open question whether constraining chicks to rely specifically on abstract numerosity, by controlling for continuous variables that typically covary with it, would lead to a different strategy or behavioural outcome.

Overall, by successfully replicating both the preference for the larger familiar set after imprinting and the preference for a prime set following habituation to even sets, we provide compelling evidence that quantity discrimination emerges very early in development. Our findings support the view that different mechanisms, emerging at the very onset of life, are likely engaged, triggering two different behaviours in the baby chicks. Imprinting to identical objects appears to trigger processes related to social affiliation and re-joining, leading chicks to favour the larger set of familiar objects. From an

evolutionary perspective, this tendency may be advantageous, as recognising and approaching a larger group of social companions can support brood cohesion, reduce predation risk through aggregation, and improve opportunities for foraging and information transfer within the group. Numerical cognition thus plays a key role in group dynamics and constitutes a foundation for ecological success in many social species, including the domestic chicken [43]. Notably, previous work has also shown that when chicks are reared with objects differing in colour, size, and shape, and are subsequently tested with completely novel objects, they express a preference toward the familiar, rather than larger, numerosity [20]. This evidence highlights how stimulus composition and the representational format available during early experience can modulate the outcome of imprinting, reinforcing the view that early processing of numerosity depends critically on how perceptual and motivational systems are structured by experience.

For what concerns habituation, exposure to visually variable stimuli seems to elicit a novelty-based strategy grounded in perceptual grouping. When imprinting-related social mechanisms are not engaged, chicks may rely more strongly on exploratory processes that favour slight novelty. Such a tendency is also adaptive in natural environments, as it enables animals to detect and respond to subtle changes in their surroundings while maintaining contact with familiar and safe stimuli. Preferring slight, rather than extreme, novelty may therefore represent a balanced strategy that promotes learning and environmental updating without abandoning potentially protective cues [37,44]. Whether these strategies are antagonistic, or whether chicks can flexibly deploy them depending on context remains an open question that requires dedicated studies.

## Supporting information

**S1 Dataset. Distinct early-life mechanisms of quantity discrimination in domestic chicks.**
(XLSX)

## Author contributions

**Conceptualization:** Maria Loconsole, Lucia Regolin.

**Data curation:** Maria Loconsole.

**Formal analysis:** Maria Loconsole.

**Funding acquisition:** Lucia Regolin.

**Investigation:** Maria Loconsole, Elisa Tedaldi.

**Methodology:** Maria Loconsole, Lucia Regolin.

**Project administration:** Lucia Regolin.

**Resources:** Lucia Regolin.

**Supervision:** Lucia Regolin.

**Validation:** Maria Loconsole, Elisa Tedaldi, Lucia Regolin.

**Visualization:** Maria Loconsole.

**Writing – original draft:** Maria Loconsole.

**Writing – review & editing:** Elisa Tedaldi, Lucia Regolin.

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
