## [Decision Letter · Decision Letter 0]

4 Mar 2026

PONE-D-26-05031Distinct early-life mechanisms of numerical discrimination in domestic chicksPLOS One

Dear Dr. Loconsole,

Thank you for submitting your manuscript to PLOS ONE. After careful consideration, we feel that it has merit but does not fully meet PLOS ONE’s publication criteria as it currently stands. Therefore, we invite you to submit a revised version of the manuscript that addresses the points raised during the review process.

We look forward to receiving your revised manuscript.

Kind regards,

Lesley Joy Rogers, B.Sc. (Hons), D.Phil., D.Sc.

Academic Editor

PLOS One

Journal Requirements:

2. To comply with PLOS One submissions requirements, in your Methods section, please provide additional information regarding the experiments involving animals and ensure you have included details on (1) methods of sacrifice, (2) methods of anesthesia and/or analgesia, and (3) efforts to alleviate suffering.

“Not applicable”

7. We note that Figures 1 and 2 in your submission contain copyrighted images. All PLOS content is published under the Creative Commons Attribution License (CC BY 4.0), which means that the manuscript, images, and Supporting Information files will be freely available online, and any third party is permitted to access, download, copy, distribute, and use these materials in any way, even commercially, with proper attribution. For more information, see our copyright guidelines: http://journals.plos.org/plosone/s/licenses-and-copyright.

1. You may seek permission from the original copyright holder of Figure(s) [#] to publish the content specifically under the CC BY 4.0 license.

Additional Editor Comments:

Please address all of the reviewers’ comments.

Reviewers' comments:

Reviewer's Responses to Questions

**Comments to the Author**

1. Is the manuscript technically sound, and do the data support the conclusions?

Reviewer #1: Yes

Reviewer #2: Partly

2. Has the statistical analysis been performed appropriately and rigorously? 

Reviewer #1: Yes

Reviewer #2: Yes

3. Have the authors made all data underlying the findings in their manuscript fully available?

Reviewer #1: Yes

Reviewer #2: Yes

4. Is the manuscript presented in an intelligible fashion and written in standard English?

Reviewer #1: Yes

Reviewer #2: Yes

5. Review Comments to the Author

Reviewer #1: In this manuscript, the authors test whether two apparently divergent findings in the literature on early numerical cognition in domestic chicks can be replicated under closely aligned methodological conditions. One line of research has shown that chicks imprinted on a set of identical objects preferentially approach the larger numerosity at test. Another line has demonstrated that chicks habituated to heterogeneous even-numbered sets display longer inspection of prime-numbered sets, suggesting reliance on perceptual grouping rather than magnitude per se. To directly compare these phenomena, the authors tested two independent groups of 24-hour-old chicks using the same numerical comparison (5 vs. 9), the same arena, and the same dependent measure (time spent in proximity to each set). They report a significant interaction between numerosity and pre-test procedure: imprinted chicks prefer the larger set (9), whereas habituated chicks spend more time near the prime set (5), replicating both previously reported effects under aligned conditions.

This is a well-designed, methodologically careful, and clearly written study. The statistical approach is appropriate and transparent, inter-rater reliability is strong, and the attempt to equate key methodological parameters across paradigms is particularly commendable. Importantly, the manuscript contributes to the literature by conducting a comparative replication of two apparently opposed effects, showing that both are robust and replicable when tested under similar experimental constraints. This comparative approach substantially strengthens confidence in the existence of multiple early-life mechanisms supporting numerical discrimination in chicks. I therefore strongly recommend publication, after minor revisions aimed at strengthening the theoretical framing and clarifying certain conceptual distinctions.

Detailed comments

1. Expand the discussion on why the two procedures elicit different mechanisms. While the authors successfully demonstrate that imprinting leads to a preference for the larger familiar set and habituation leads to a preference for the prime set, the discussion would benefit from a slightly more explicit consideration of why these procedures might differentially engage distinct cognitive systems.

2. Situate results within imprinting studies using heterogeneous stimuli. The discussion would benefit from acknowledging that not all imprinting paradigms lead to a preference for the larger numerosity. Previous work has shown that when chicks are imprinted on heterogeneous objects, they may exhibit a preference for the familiar numerosity, rather than the larger one.

This literature is highly relevant here because it suggests that imprinting does not uniformly trigger a magnitude-based strategy; instead, stimulus composition and representational format can modulate the outcome. Integrating these findings would reinforce the manuscript’s broader claim that early numerical behavior depends critically on how experience structures perceptual and motivational systems.

In this regard, it is important to note that the stimuli used during habituation cannot be straightforwardly classified as either homogeneous or heterogeneous in terms of shape. Within each individual set, elements were homogeneous (all items shared the same shape and colour), but every 10 seconds a new set composed of different shapes and numerosities was presented, resulting in an overall visually heterogeneous experience across the habituation phase.

3. Clarify the conceptual distinction between imprinting and habituation. The manuscript treats imprinting and habituation as distinct pre-test procedures, which is experimentally accurate. However, from a developmental perspective in precocial species such as the domestic chick, this distinction may not be entirely straightforward. We know that imprinting learning may begin after relatively short exposure durations and that, especially short imprinting periods can give rise to novelty preferences (see the old works of Patrik Bateson on this regard). Given this, it would be useful for the authors to clarify on what theoretical basis they consider the habituation condition to engage mechanisms fundamentally distinct from imprinting learning.

Reviewer #2: In this manuscript, the authors replicate two experiments on numerosity in young chicks (choice of the larger group after imprinting, and preference for the “prime” number following habituation to even numerosities), with the aim of standardizing the setup and experimental conditions. While this objective is valuable and worth pursuing, several issues limit the strength of the claims.

1) Lack of true standardization across experiments

The stated goal of standardizing conditions across the two experiments is not fully achieved. Although the authors employ a common physical setup and the same test numerosities (5 vs 9), the stimuli themselves differ substantially.

In Experiment 1, physical red cylinders are used, whereas in Experiment 2 stimuli are digitally projected on a screen and vary in both color and shape. This discrepancy represents a significant limitation for two main reasons: 1) It weakens the claim of methodological standardization across experiments; 2) It limits the degree of stimulus control, particularly in Experiment 1 (see point 2).

Designing Experiment 1 using digitally controlled stimuli, as in Experiment 2, would have allowed for tighter control of continuous variables and improved comparability between experiments. It is unclear why this issue is neither acknowledged nor discussed.

2) Insufficient control and discussion of non-numerical continuous variables

Despite the title emphasizing numerical discrimination, the manuscript does not adequately address continuous variables that chicks might use instead of number (e.g., total area, perimeter, density, spatial arrangement, convex hull). This omission substantially weakens the interpretation of the results: without explicit control or analysis of these factors, it remains unclear whether the observed effects genuinely reflect numerical processing.

This concern is particularly relevant for Experiment 1, where the use of physical objects makes precise control of continuous variables more difficult. This further supports the argument that digital stimuli would have been more appropriate.

Regarding Experiment 2, although digital stimuli were used, there is no discussion of their geometrical properties or whether continuous variables covaried with number and potentially contributed to the results. Such analyses are essential.

A further issue concerns the habituation phase. During habituation, chicks were exposed to numerosities 4, 6, 10, and 12, each apparently associated with a specific color and shape (the manuscript should clearly specify which color/shape corresponded to each numerosity). Importantly 4 is smaller than both test numerosities (5 and 9); 6 lies between them, 10 and 12 are larger than both.

It would be important to test whether choices at test are modulated by the specific color/shape previously associated with different numerical magnitudes. An analysis stratified by habituation category (smaller, intermediate, larger than test values) could clarify whether the observed effects depend on prior magnitude exposure rather than numerical properties per se.

3) Theoretical framework of “prime number” choice

There are conceptual concerns regarding the interpretation of results in terms of prime number detection.

First, the test condition (5 vs 9) alone is insufficient to support a strong claim about prime number preference; additional testing pairs would be necessary to substantiate such a conclusion.

Second, although the rationale is based on Loconsole et al., 2021, the theoretical background is not adequately explained in the current manuscript. The explanation is brief and unclear. In the earlier work, the emphasis appears to be on symmetry properties and grouping mechanisms rather than on prime numbers per se. The novelty or asymmetry of grouping patterns seems central to the interpretation.

The current manuscript risks overstating “prime number detection” without sufficiently grounding it in the symmetry/grouping framework. This conceptual framing should be clarified and expanded.

Moreover, if the proposed mechanism relies on symmetry properties, then the spatial arrangement and configuration of elements become critical factors. A deeper analysis of stimulus configuration and grouping structure would strengthen the interpretation.

Overall assessment and recommendations

Substantial revision is necessary. In particular:

• Experiment 1 should ideally be redesigned using digitally controlled stimuli to match Experiment 2 and to allow systematic control of continuous variables (e.g., varying area, perimeter, density, convex hull during imprinting to prevent their use as informative cues).

• Experiment 2 should include analyses assessing the contribution of continuous variables that covary with number.

• Test performance should be analyzed as a function of the color/shape associated with specific habituation numerosities.

• The discussion should be substantially revised to address these methodological issues and to more accurately frame the concept of prime number detection within the broader context of symmetry and grouping mechanisms.

Without these adjustments, the conclusions regarding numerical discrimination and prime number preference remain insufficiently supported.

6. PLOS authors have the option to publish the peer review history of their article (what does this mean?). If published, this will include your full peer review and any attached files.

Reviewer #1: No

Reviewer #2: No

---

## [Author Response · Author response to Decision Letter 1]

23 Mar 2026

A detailed response to the Reviewers' comments is available in the uploaded file Response_to_review_Loconsole

---

## [Decision Letter · Decision Letter 1]

22 Apr 2026

PONE-D-26-05031R1Distinct early-life mechanisms of numerical discrimination in domestic chicksPLOS One

Dear Dr. Loconsole,

Thank you for submitting your manuscript to PLOS ONE. After careful consideration, we feel that it has merit but does not fully meet PLOS ONE’s publication criteria as it currently stands. Therefore, we invite you to submit a revised version of the manuscript that addresses the points raised during the review process.

We look forward to receiving your revised manuscript.

Kind regards,

Lesley Joy Rogers, B.Sc. (Hons), D.Phil., D.Sc.

Academic Editor

PLOS One

Journal Requirements:

Additional Editor Comments (if provided):

Your manuscript requires just some minor revision, as indicated by Reviewer 2. I look forward to seeing it again.

Reviewers' comments:

Reviewer's Responses to Questions

**Comments to the Author**

1. If the authors have adequately addressed your comments raised in a previous round of review and you feel that this manuscript is now acceptable for publication, you may indicate that here to bypass the “Comments to the Author” section, enter your conflict of interest statement in the “Confidential to Editor” section, and submit your "Accept" recommendation.

Reviewer #1: All comments have been addressed

Reviewer #2: (No Response)

2. Is the manuscript technically sound, and do the data support the conclusions?

Reviewer #1: Yes

Reviewer #2: Partly

3. Has the statistical analysis been performed appropriately and rigorously? 

Reviewer #1: Yes

Reviewer #2: Yes

4. Have the authors made all data underlying the findings in their manuscript fully available?

Reviewer #1: Yes

Reviewer #2: Yes

5. Is the manuscript presented in an intelligible fashion and written in standard English?

Reviewer #1: Yes

Reviewer #2: Yes

6. Review Comments to the Author

Reviewer #1: (No Response)

Reviewer #2: I thank the authors for addressing all my concerns. Overall, I am satisfied with the explanations and their implementation in the manuscript. However, I still believe that a stronger emphasis on these points is needed to avoid any misleading initial interpretation.

I outline my response to the three main points below:

• Physical stimuli in Experiment 1. I agree with the authors that physical stimuli can be more engaging than on-screen digital stimuli for imprinting. However, there is substantial evidence, including studies cited in the Introduction, as well as previous work with naïve chicks in numerical discrimination (e.g., Lorenzi et al., 2024; Helyon), showing that digital stimuli are also effective. Anyway, the current discussion is acceptable, particularly in light of the clarification that the study targets the perception of quantitative information (with chicks potentially relying on non-numerical cues, as occurs naturally).

• Control of non-numerical cues. The authors explain that controlling these cues was not their aim, as they sought to approximate natural conditions and study magnitude encoding. This rationale is valid, but the short paragraph added to clarify it seems insufficient to me. It should be stated clearly from the outset, with an explicit distinction between numerical cognition and more general quantitative cognition. I suggest first introducing a clear terminological clarification in the Introduction, distinguishing between numerical cognition and other magnitude systems, as standard in the literature.

After introducing this distinction, the authors should explicitly state that their aim is to study general quantity discrimination, where judgments may rely on numerical information as well as other cues (e.g., area, density). To avoid confusion, I strongly suggest then avoiding the terms numerosity and numerical throughout the manuscript, in favor of more general terms such as quantity or quantitative. For the same reason, the title (as well as the abstract) should be revised to avoid “numerical discrimination” term, which is not precise with respect to the experiments’ outcomes. Similarly, statements in the Discussion (e.g., line 255) could be rephrased to improve precision.

• “Prime numbers”. I am satisfied with the authors’ response and their clarification that the effect likely relies on perceptual properties that may also characterize prime numbers. This explanation should be emphasized in the manuscript. In particular, the presentation should avoid an initial focus on prime numbers, which can be misleading. The task should instead be framed primarily in terms of perceptual grouping ability, with any reference to prime numbers introduced only as a secondary interpretation in the Discussion. Along the same lines, I suggest avoiding the term “prime-numbered set” in the abstract when presenting the results. In the Introduction, lines 70–81 could be rephrased to refer to perceptual grouping discrimination rather than prime numbers.

Overall, while I agree with the authors’ clarification, I strongly encourage them to integrate these considerations still more thoroughly throughout the manuscript to improve clarity and precision of interpretation.

Minor comments:

• Some references are reported in an inconsistent style (e.g., lines 269–270).

7. PLOS authors have the option to publish the peer review history of their article (what does this mean?). If published, this will include your full peer review and any attached files.

Reviewer #1: No

Reviewer #2: No

---

## [Author Response · Author response to Decision Letter 2]

4 May 2026

We thank the Editor and the Reviewers for their thorough revision of our manuscript. We have provided a detailed point-by-point to the Reviewers observations in the Response to Reviewers file.

---

## [Editor Report · Decision Letter 2]

6 May 2026

Distinct early-life mechanisms of quantity discrimination in domestic chicks

PONE-D-26-05031R2

Dear Dr. Loconsole,

We’re pleased to inform you that your manuscript has been judged scientifically suitable for publication and will be formally accepted for publication once it meets all outstanding technical requirements.

Kind regards,

Lesley Joy Rogers, B.Sc. (Hons), D.Phil., D.Sc.

Academic Editor

PLOS One

Additional Editor Comments (optional):

Thank you for addressing the reviewer's second set of comments. Your manuscript is now much improved and ready for publication. It adds important information to the field.
---

## [Editor Report · Acceptance letter]

PONE-D-26-05031R2

PLOS One

Dear Dr. Loconsole,

I'm pleased to inform you that your manuscript has been deemed suitable for publication in PLOS One. Congratulations! Your manuscript is now being handed over to our production team.

Kind regards,

on behalf of

Prof. Lesley Joy Rogers

Academic Editor

PLOS One